# The Protective Effect of Sericin on AML12 Cells Exposed to Oxidative Stress Damage in a High-Glucose Environment

**DOI:** 10.3390/antiox11040712

**Published:** 2022-04-05

**Authors:** Feng-Ya Jing, Yu-Jie Weng, Yu-Qing Zhang

**Affiliations:** School of Biology & Basic Medical Sciences, Soochow University, Suzhou 215123, China; 20204221001@stu.suda.edu.cn (F.-Y.J.); 20184021002@stu.suda.edu.cn (Y.-J.W.)

**Keywords:** sericin, degumming, antioxidant, oxidative stress, antihyperglycemic effect, inflammatory factor

## Abstract

Two types of sericin peptides with high molecular weight (HS) and low molecular weight (LS) were obtained by the green water boiling ultrasonic method and the Ca(OH)_2_ ultrasonic method, respectively. In this experiment, a high-glucose medium was used to simulate a high-glucose environment in the body, and appropriate concentrations of normal alpha mouse liver 12 (AML12) hepatocytes were exposed to a series of concentrations of HS and LS. The effects of the two sericin peptides on AML12 cells in a high-glucose environment were investigated in detail in terms of oxidative stress and inflammatory factor expression in cells. HS and LS-groups reduced the levels of oxidative stress, inflammation, and tumor necrosis factor (TNF), and the latter significantly reduced the levels of TNF-α, interleukin (IL)-6, and nuclear factor (NF)-κB in AML12 cells. Additionally, it significantly reduced the oxidative stress damage caused by the high-glucose environment compared with normal AML12 cells. These results indicate that sericin may be an antioxidant recovered from industrial waste, and has potential and for use in the reduction of environmental pollution and the development of functional foods with antioxidation and antihyperglycemic effect.

## 1. Introduction

The liver is the predominant location of glucose metabolism in the body. Other cells use glucose in cellular respiration, which produces a large amount of energy to maintain cell metabolism. Hepatocytes integrate glucose and store it in the form of glycogen. Liver cells contain large amounts of endoplasmic reticulum, where lipid synthesis, protein folding, and transportation activities occur frequently [1]. A growing body of research has confirmed that endoplasmic reticulum stress potentially impacts inflammation and metabolic diseases. Oxidative stress (OS) is caused by the combination of excessive oxygen free radicals and insufficient antioxidants. Increased OS can destroy the oxidation state and redox potential in cells; further, it can lead to liver cell damage and apoptosis [2,3,4]. In diabetic rats, OS and inflammation induced by streptozotocin (STZ) are greatly increased [5]. At the cellular level, it can also enhance endoplasmic reticulum stress and cause apoptosis. More and more evidence indicates that metabolic syndrome caused by oxidative stress may also be related to the overproduction of cellular inflammatory factors, including IL-1 and IL-18 [6,7].

Many amino acids are effective antioxidants [8,9,10,11,12]. In addition, several natural macromolecules, especially high-molecular-weight proteins, protect against oxidative stress [13,14,15]. Among these, sericin may serve an interesting role as an antioxidant. Sericin, an active biomolecule, is a polypeptide containing 18 amino acids; most of these amino acids have strong polar side groups, such as hydroxyl, carbonyl, and amino groups [16], and are rich in serine [17], which can inhibit lipid peroxidation [18] and promote fibroblast attachment and growth [19]. It also has a variety of pharmacological and therapeutic effects [20,21,22,23]. Sericin inhibits ultraviolet B (UVB)-induced acute damage and antitumor effects by reducing oxidative stress in the skin of nude mice [24]. In addition, it provides a cryopreservation protection effect on cells [25], and adding a culture medium can promote cell proliferation [26]. Dietary sericin can reduce the fat levels in rats fed a high-fat diet and improve their glucose tolerance [27], reduce oxidative stress, and promote cell proliferation and nitric oxide production [28]. After skin fibroblasts were precultured with sericin, the oxidative stress induced by hydrogen peroxide was reduced, and cell viability was almost restored to the level of the control group. The activities of catalase, lactate dehydrogenase, and malondialdehyde in the cell were significantly reduced, demonstrating the antioxidative stress effect of sericin [29]. In vitro experiments revealed that sericin can scavenge multiple reactive oxygen species (ROS), and it can have anti-tyrosinase, anti-elastase, and immunomodulatory effects [30].

In recent years, our group has developed a method for green degumming with calcium hydroxide and the complete recovery of low-molecular-weight sericin peptides and their hydrolysates. This water-soluble sericin is easily absorbed and utilized by the human body. In vitro studies have shown that it has a certain degree of resistance against oxidation and free radical scavenging activity; the IC_50_ values for tyrosinase and α-glucosidase are 3.5 mg/mL and 8.5 mg/mL, respectively [31]. The models of type 2 diabetic mice and rats are constructed by the injection of STZ. Oral administration for 4 weeks significantly alleviated hyperglycemia in diabetic mice, and abnormal glucose tolerance and insulin resistance of diabetic mice improved. Abnormally elevated glycosylated serum protein; serum liver function enzyme activity; and the levels of triglyceride (TG), low-density lipoprotein cholesterol (LDL-C), malondialdehyde (MDA), and 8-hydroxydeoxyguanosine (8-OHdG) were also reduced in model mice. The levels of high-density lipoprotein cholesterol (HDL-C), catalase (CAT), and total superoxide dismutase (T-SOD) increased significantly in diabetic mice. These results demonstrated that it can regulate lipid metabolism in type 2 diabetic mice [32,33]. However, in-depth research on the effect of this small-molecular-weight sericin peptide on oxidative stress and inflammatory factors at the cellular level is still lacking.

Normal alpha mouse liver 12 (AML12) cells, a type of liver parenchymal cell, are a common and well-established hepatocyte model. AML12 cells have a typical polygonal epithelial phenotype and serve as an adequate model for the effects of liver oxidative stress, lipid peroxidation, and inflammation. In this experiment, a high-sugar medium was used to imitate the formation of a high-sugar environment in the body, and various concentrations of AML12 cells were exposed to a series of concentrations of HS and LS. Oxidative stress and inflammatory factor expression in these cells were investigated in detail.

## 2. Materials and Methods

### 2.1. Materials

A CELL 150i CO_2_ incubator was obtained from Mettler Toledo Instrument Company, and a UV spectrophotometer U-3000 was acquired from HITACHI Co. Ltd., Tokyo, Japan. A multifunctional microplate reader (MD SpectraMax M5, USA) was obtained from American Molecular Devices Co., Ltd. RPMI-1640 medium, high glucose medium, fetal bovine serum, the bicinchoninic acid (BCA) protein concentration determination kit, and the cell counting kit 8 (CCK8) were acquired from Shenggong Bioengineering Co., Ltd., Shanghai, China. ELISA kits of T NF-α (Catalog No. PI328) and IL-6 (Catalog No. PT516) were obtained from Beyotime Biotechnology Co., Ltd. (Shanghai, China). The NF-κB ELISA kit (Catalog No. F16343) was purchased from Westang Biotechnology Co., Ltd. (Shanghai, China).

### 2.2. Sericin Sample Preparation

After an appropriate amount of the clean cocoon shells of silkworm *Bombyx mori* was weighed, the preparation of the two sample powders of low-molecular-weight sericin (LS) and high-molecular-weight sericin (HS) was carried out by the method just published in the journal by our research group [34]. The molecular weight distribution range of this degraded sericin peptide and its hydrolyzates is usually less than 25 kDa. The distribution range of the HS samples obtained by water cooking is very wide, and the molecular weight is still very large and often less than 300 kDa. These two sericin samples can also usually be identified using the SDS-PAGE method previously reported by our group [35,36].

### 2.3. Cell Culture

This experiment examined the effects of two kinds of sericin, HS and LS, on the cell culture. Twelve groups were assessed in the experiment, including the control group (normal medium), the high-glucose culture or model group (RPMI-1640 medium + 30 mmol/L of glucose), and the sample group (RPMI-1640 medium + 30 mmol/L glucose + final concentration of 30, 60, 90, 120, or 150 μg/mL of sericin samples). The specific experimental procedure follows the cell culture method previously reported elsewhere [37].

### 2.4. Cell Viability Determination

The murine hepatocyte cell line AML12 was purchased from the Procell Life Science & Technology Co., Ltd. (Wuhan, China). Cells were cultured in a RPMI-1640 medium supplemented with 10% fetal bovine serum (FBS) and 100 units/mL penicillin/streptomycin mixture (Gibco^TM^, NY, USA). The AML12 cells in the log phase were prepared into a suspension, and an appropriate amount of cell suspension (approximately 7 × 10^3^ cells) was added to the 48-well plate. This cell was incubated for 24 h at 37 °C in 5% CO_2_. A final concentration of 30~150 μg/mL of samples were added, and incubation was continued for 24 h. Then, 10 μL of CCK8 was added to each well, the plate was incubated at 37 °C for 1 h, and the absorbance was measured [38]. The detailed experimental procedure was performed with the cell culture method previously reported by Wang et al. [37].

### 2.5. Cellular ROS Level Measurement

Cellular reactive oxygen species (ROS) levels were determined by the chemofluorescence method with an ROS assay kit purchased from the Nanjing Jiancheng Bioengineering Institute (Nanjing, China). The DCFH-DA (2,7-dichlorofuorescin diacetate) probe used in this kit is by far the most commonly used and sensitive probe for the detection of intracellular reactive oxygen species. The experimental procedure followed the cell culture method previously reported by Wang et al. [37] A suspension of AML12 cells in the log phase was prepared, and 2 mL (approximately 1 × 10^6^ cells) was added to each well of a six-well plate, which was incubated for 48 h at 37 °C in 5% CO_2_. A final concentration of 30~150 μg/mL of samples were added, and the culture was continued for 24 h. The subsequent steps were performed following the instructions of the ROS Determination Kit (Chemifluorescence) from Nanjing Jiancheng Co., Ltd., Nanjing, China.

### 2.6. Immune Damage Factor Level Determination

The determination of immune-damaging factors mainly refers to the previously reported methods by authors with slight modifications [39,40,41]. A suspension of AML12 cells in the log phase was prepared, 200 μL (approximately 5 × 10^5^ cells) was added to each well of the six-well plate, and sterile PBS was added to the edge holes. The plate was incubated in 5% CO_2_ at 37 °C for 24 h. A final concentration of 30~150 μg/mL of samples were added, and incubation was continued for 24 h. Each group of cells was collected and merged. When the cell suspension was diluted with PBS, the supernatant was directly used to measure the levels of the first two inflammatory factors IL-6 and TNF-α [42,43], whereas the cell suspension for the determination of NF-κB level was diluted with PBS, the cells were destroyed, and the intracellular components were released through repeated freezing and thawing [44,45]. After centrifuging at 7200× *g* for 20 min, the supernatant was collected.

Standard holes, blank holes, and sample holes were set for sample measurement. Standards of different concentrations were added to the standard wells, 10 μL of the sample was added to the sample wells to be tested, and 40 μL of sample diluent was added. After incubating at 37 °C for 30 min, the plate was washed, 50 μL of enzyme-labeled reagent was added, and the plate was incubated again at 37 °C for 30 min. The plate was then washed, 50 μL of color reagents A and B were added, and the plate was incubated at 37 °C for 15 min. Finally, 50 μL of stop solution was added, and the absorbance of each hole with a blank hole was measured at 450 nm.

In the above experiment, the minimum detection amounts of rat TNF-α and rat IL-6 ELISA kits were 26 pg/mL and 67 pg/mL, respectively. The two ELISA Kits specifically, sensitively, and quantitatively detected TNF-α and IL-6 levels in rat serum, plasma, cell or tissue lysate, or cell culture supernatant. The minimum detection concentration of NF-κB ELISA kit was less than 15 pg/mL, which can specifically detect recombinant or natural rat NF-κB and does not cross-react with other rat cytokines.

In this test, the BCA protein concentration kit (purchased from Shanghai Biyuntian Biotechnology Co., Ltd., Shanghai, China) was used to determine the protein concentration.

### 2.7. Data Treatment

The experimental data are expressed as the mean ± standard deviation (±SD, *n* = 5). Significant differences between two sets of data were assessed using one-way analysis of variance (ANOVA, Origin 8.5 version). A value of *p* < 0.05 was statistically significant.

## 3. Results

### 3.1. Protective Effects of Oxidative Stress Damages

The murine hepatocyte cell line AML12 is derived from the liver of transgenic mice overexpressing a transforming growth factor α (TGFα) [46] and has mainly been used for studies on lipid metabolism and liver injury [40,46]. Here, the damage to AML 12 cells under a high glucose environment is mainly characterized. After AML12 cells were stimulated by 30 mmol/L glucose, the protective effects of the two sericins on the oxidative stress of the cells in a high-glucose environment were investigated. In this experiment, the CCK8 method was used to assess cell proliferation and toxicity. As shown in Figure 1, as the concentration of sericin increased, the viability of AML12 cells gradually improved. Under the same treatment concentration, the protective effect of LS on the viability of AML12 cells was superior to that of HS. When the sericin concentration increased to 150 μg/mL, the cell viability of AML12 reached the highest value. This demonstrates that sericin, especially the LS prepared using the Ca(OH)_2_-ultrasonic degumming method, has a strong protective effect against oxidative stress in AML12 cells in a high-glucose environment.

### 3.2. ROS Level in Cells

ROS include peroxides, superoxides, hydroxyl radicals, and other oxidative substances, which play key roles in cell signal transduction and homeostasis. The high-glucose environment can lead to increased ROS levels, which can cause serious damage to the cell structure and cell oxidative stress damage. Figure 2 demonstrates that the ROS level of AML12 cells in the high-glucose treatment group was significantly higher than that in the normal group (*p* < 0.01), which was 1.6 times that of the normal group. After adding two different concentrations of sericin, intracellular ROS levels decreased significantly. Comparing the two sericins, the LS group reduced the level of ROS in AML12 cells more significantly. With increased HS concentration, the intracellular ROS level of AML12 in the HS group gradually decreased, and the intracellular ROS level of AML12 under the 150 μg/mL HS treatment was only 98% of that of the normal group. After co-incubation with LS (30–150 μg/mL), the ROS level in AML12 cells returned to normal levels. This demonstrates that both sericins, especially LS, can reduce the level of cellular oxidative stress in a high-glucose environment.

### 3.3. TNF-α Level in Cells

TNF-α was first discovered by researchers in macrophages, which can induce hemorrhagic necrosis of tumor tissues and thereby inactivate tumor cells. TNF-α is closely related to immune regulation and inflammatory response and can be used to evaluate drug treatment efficacy. As shown in Figure 3, the TNF-α level in AML12 cells of the normal group was 201.13 ± 4.73 ng/mg prot. After high glucose induction, TNF-α in AML12 cells increased significantly to 574.04 ± 13.96 ng/mg prot, which was 1.87 times higher than the level in the normal group. After adding two different concentrations of sericin, the intracellular TNF-α content was significantly reduced, and both exhibited obvious dose-dependent effects. Compared with the two sericins, the LS group reduced the content of TNF-α in AML12 cells more significantly. As HS concentration increased, the TNF-α content in the AML12 cells of the HS group gradually decreased, and the TNF-α content in AML12 cells treated with 150 μg/mL HS was 214.24 ± 17.5 ng/mg prot, returning to normal levels. With increased LS concentration, the TNF-α content in the AML12 cells of the LS group decreased significantly, and the TNF-α contents in the AML12 cells exposed to 120 and 150 μg/mL LS treatments were 197.06 ± 13.1 ng/mg prot and 161.83.06 ± 16.1 ng/mg prot, respectively, which were lower than normal levels. This indicates that the two sericins, especially LS, may slow down the inflammatory response of AML12 cells and even in the entire body by lowering TNF-α levels.

### 3.4. IL-6 Levels in Cells

IL-6 is produced by T cells and fibroblasts. It is involved in regulating the proliferation and differentiation of bone marrow stem cells, affecting the body’s hematopoietic function and immune response; it also has anti-infection and anti-inflammatory effects. As shown in Figure 4, the IL-6 content in the AML12 cells in the normal group was 58.23 ± 1.48 pg/mg prot. After high-glucose induction, IL-6 in AML12 cells increased significantly to 182.41 ± 6.16 pg/mg prot, which was 2.13 times higher than the level of the normal group. After adding two different concentrations of sericin, the intracellular IL-6 content was significantly reduced, and both concentrations exhibited obvious dose-dependent effects. Comparing the two sericins, the LS group lowered the IL-6 content in the AML12 cells more significantly. With increased HS concentration, the IL-6 content in the AML12 cells of the HS group gradually decreased, and the IL-6 content in the AML12 cells exposed to 120 μg/mL HS treatment was 62.24 ± 2.93 pg/mg prot, returning to normal levels. The IL-6 content of the AML12 cells treated with 150 μg/mL HS was 44.99 ± 0.93 pg/mg prot, which were lower than normal levels. With increased LS concentration, the IL-6 content in the AML12 cells of the LS group decreased significantly, and the IL-6 contents in the AML12 cells under 90, 120, and 150 μg/mL LS treatments were 56.58 ± 2.14 pg/mg prot, 46.29 ± 1.23 pg/mg prot, and 40.67 ± 1.39 pg/mg prot, respectively, which were lower than normal levels. This indicates that the two sericins, especially the addition of >90 μg/mL LS, may significantly reduce the inflammatory response of the AML12 cells and in the entire body by lowering the IL-6 level.

### 3.5. NF-κB Level in Cells

NF-κB is a DNA-protein complex widely present in animal cells, which can regulate DNA transcription and affect the cell’s response to external factors, such as inflammatory factors. NF-κB participates in infection response, immunity, development, cancer suppression, and other activities. As shown in Figure 5, the content of NF-κB in AML12 cells of the normal group was 172.83 ± 13.31 ng/mg prot. After high glucose induction, the NF-κB level in the AML12 cells significantly increased to 585.83 ± 8.95 ng/mg prot, which was 3.39 times higher than that of the normal group. After adding two different concentrations of sericin, the intracellular NF-κB content decreased significantly, and both concentrations showed obvious dose-dependent effects. Comparing the two sericins, the LS group reduced the content of NF-κB in AML12 cells more significantly. With increased HS concentration, the content of NF-κB in the AML12 cells of the HS group gradually decreased, and the content of NF-κB in the AML12 cells treated with 120 μg/mL HS was 179.23 ± 8.93 ng/mg prot, returning to normal levels. The content of NF-κB in AML12 cells treated with 150 μg/mL HS was 152.39 ± 9.51 ng/mg prot, which were lower than normal levels. With increased LS concentration, the NF-κB content in the AML12 cells of the LS group decreased significantly, and the NF-κB contents in the AML12 cells under the levels of 120 and 150 μg/mL LS treatments were 144.15 ± 13.85 and 124.8 ± 0.21 ng/mg prot, respectively, which were lower than normal levels. This indicates that the two sericins may slow down the inflammatory response of the AML12 cells and in the entire body by lowering NF-κB.

## 4. Discussion

There are several layers of sericin wrapped around silk fibers. Their amino acid composition is not exactly the same, but they are all polar amino acids in the majority. At present, there are many reports on the preparation and recovery methods of sericin. The resulting sericins have large differences in its molecular weight distribution, resulting in large differences in biological activities, especially antioxidant activities. There have been many reports on the antioxidant activity of sericin [29,30,47,48,49,50,51,52,53], and the authors have also reviewed and discussed its function and application [54,55]. Recently, our team’s investigation confirmed that co-incubating cells with sericin can effectively promote cell proliferation [56]. In the culture experiment of sericin as a serum substitute, different types of cells have different proliferation effects. In the proliferation test of four types of tumor cells (BEL-7402, A549, MCF-7, and HepG-2), the sericin enzymatic hydrolysates can ensure the complete replacement of animal serum within 0–4 days [26]. Mouse fibroblasts (L929) grow and proliferate faster in a medium containing 0.75% sericin hydrolysate. Zhao et al. also found that the alcohol extract of sericin from the green cocoon layer is not only non-cytotoxic to human liver cells (L02), but it also protects growth and proliferation in a high-sugar environment [40].

More recently, the author’s team have again developed a novel green preparation method of sericin [36]. The resultant sericin peptide and its hydrolyzate have good water solubility and can be added to food and utilized as functional health food [31]. This sericin peptide and its hydrolyzate are called as low-molecular-weight sericin (LS). The in vitro antioxidative activity of LS have been published in the latest *Biomolecules* [34].

In this experiment, a high-glycemic medium containing 30 mmol/L glucose was used to mimic the formation of a high-glycemic environment in the body, and the AML12 cells were exposed to different concentrations of HS and LS high-glycemic media. After 24 h, using the viability of the AML12 cells without sericin as a control, the addition of both HS and LS enhanced the viability of AML12 cells in a high-glucose environment, and both exhibited a dose-dependent effect. This situation is similar to the situation in which AML12 cells are induced to produce endoplasmic reticulum stress by soft acid [57], which causes an increase in the level of intracellular ROS, leading to a lipid toxicity of AML12 cells and triggering apoptosis. In this experiment, the ROS level of AML12 cells in the high-glucose treatment group was significantly (1.6 times) higher than that in the normal group (*p* < 0.01). After adding two different concentrations of sericin, the level of intracellular ROS decreased significantly. In comparison with the two sericins, the LS group decreased the level of ROS in AML12 cells more significantly. With increased HS concentration, the intracellular ROS level of AML12 in the HS group gradually decreased, and the intracellular ROS level of AML12 exposed to 150 μg/mL HS treatment was only 98% of that of the normal group. After co-incubation with LS (30–150 μg/mL), the ROS level in AML12 cells returned to normal levels. This shows that sericin, especially the small-molecular-weight sericin prepared using the calcium hydroxide degumming method, has a strong protective effect against the oxidative stress caused by AML12 cells in a high-glucose environment.

NF-κB is an important transcription factor. Activated NF-κB can initiate the transcription of its downstream genes, leading to the overexpression of a series of inflammatory factors, including IL-6 and TNF-α. Therefore, the three inflammatory factors—NF-κB, TNF-α, and IL-6—were selected for determination. After high-glucose induction, the NF-κB level was 3.39 times higher than that of the normal group. After adding two different concentrations of sericin to the culture medium, the intracellular NF-κB level was significantly reduced, and both concentrations exhibited obvious dose-dependent effects. TNF-α can affect the breakdown of fat particles and the level of fatty acids in the body, ultimately leading to insulin resistance. The content of TNF-α in AML12 cells of the normal group was 201.13 ± 4.73 ng/mg prot. After high glucose induction, TNF-α in AML12 cells was significantly increased by 2.85 times compared with that of the normal group. After exposure to two different concentrations of sericin, the intracellular TNF-α content decreased significantly and both exhibited a significant dose-dependent effect, the ability of LS to lower the TNF-α level was more significant. TNF-α in AML12 cells treated with 150 μg/mL HS and 120 and 150 μg/mL LS returned to normal levels. IL-6 affects insulin secretion and glucose metabolism in pancreatic islets. After high glucose induction, IL-6 in AML12 cells was significantly increased by 2.13 times compared with the level in the normal group. HS and LS reduced the secretion of IL-6 in cells, and the regulation ability of LS was more significant. This demonstrates that the two types of sericin, especially LS, can slow down or reduce the inflammatory response of AML12 cells by regulating the secretion level of IL-6. The experimental results of this study are consistent with our group’s recent results, which demonstrated that oral sericin can improve the antioxidant capacity of diabetic mice, eliminate inflammatory factors, and ultimately reduce blood sugar levels [32,33].

The above experimental results all showed that LS had not only higher bioactivity and antioxidation capability, but also a higher hypoglycemic effect in T2D diabetic mice treated with oral sericin [32]. The oral experiment of HS sericin showed that its antioxidative ability in T2D diabetic rats was far less than that of LS [33], and it had little effect on lowering blood sugar (unpublished results). The investigation by Kato et al. showed that the sericin is resistant to several proteases and has a low digestibility of protein, similar to buckwheat protein. It would exert protective effects against constipation [58]. Therefore, the higher exposure of the potentially effective binding sites of the LS sample, the easier it is to enter cells; larger molecules of sericin (HS) in aqueous solution form spheres in which molecular chains have a β-folded structure [59,60], the exposure of effective binding sites is not high, it is not easy to be enzymatically hydrolyzed in vivo, and it is difficult to easily enter cells.

In the future, we will continue to use LS peptides in animal experiments to explore its mechanism of reducing blood sugar and explore new uses for recycling and green utilization of waste during the processing of this silk, especially in assisting with the development of blood sugar-lowering functional foods.

## 5. Conclusions

Sericin can reduce the oxidative stress of liver cells by scavenging superoxide free radicals. In a high-glucose environment, the viability of AML12 cells decreases, proliferation is inhibited, the level of intracellular ROS is significantly increased, and the levels of immune-damaging factors (such as TNF-α, IL-6, and NF-κB) are significantly elevated. After AML12 cells were co-incubated with 30–150 μg/mL HS and LS, especially the latter, their viability gradually increased, the level of intracellular ROS gradually decreased, and the levels of TNF-α, IL-6, and NF-κB decreased in a dose-dependent manner. LS had significantly stronger antioxidant and anti-inflammatory effects than HS. The results demonstrated that sericin can significantly reduce the oxidative stress caused by a high-glucose environment. Therefore, sericin may be an antioxidant and could have a positive effect on insulin resistance and glucose metabolism disorders.

## Figures and Tables

**Figure 1 antioxidants-11-00712-f001:**
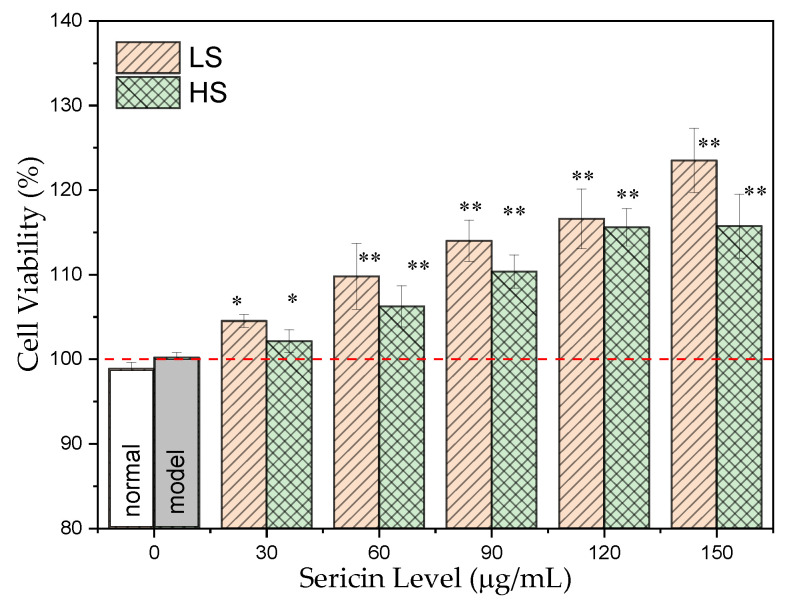
Effect of sericin peptides on the cell viability in AML12 cells cultured with high glucose. * *p* < 0.05 and ** *p* < 0.01, sample group vs. model group.

**Figure 2 antioxidants-11-00712-f002:**
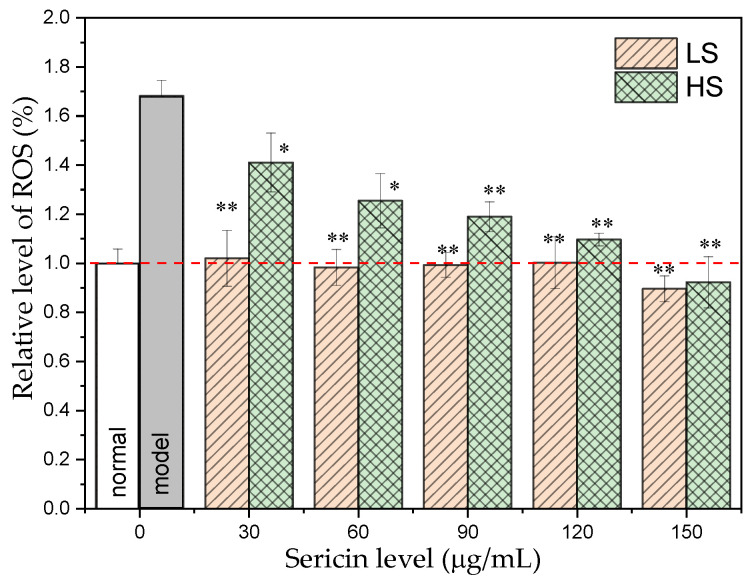
Effect of sericin peptides on ROS content in AML12 cells cultured with high glucose. * *p* < 0.05 and ** *p* < 0.01, sample group vs. model group.

**Figure 3 antioxidants-11-00712-f003:**
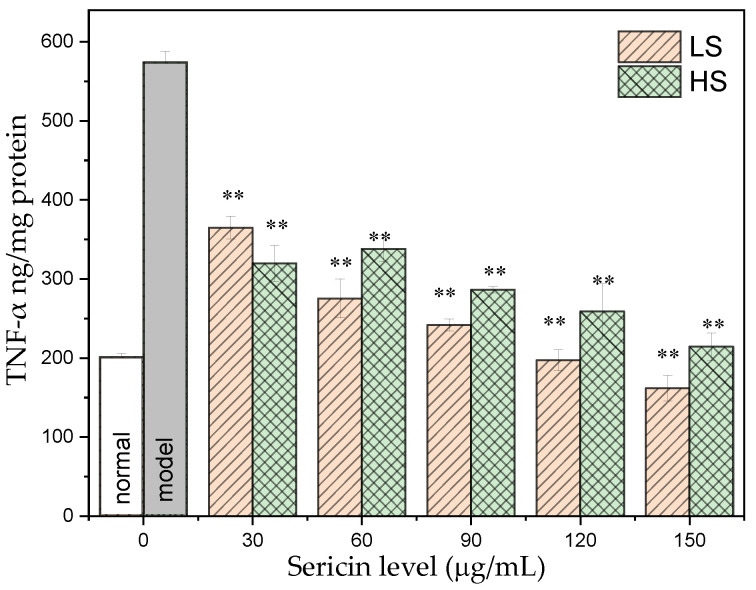
Effect of sericin on TNF-α level in AML12 cells culture with high glucose. ** *p* < 0.01, sample group vs. model group.

**Figure 4 antioxidants-11-00712-f004:**
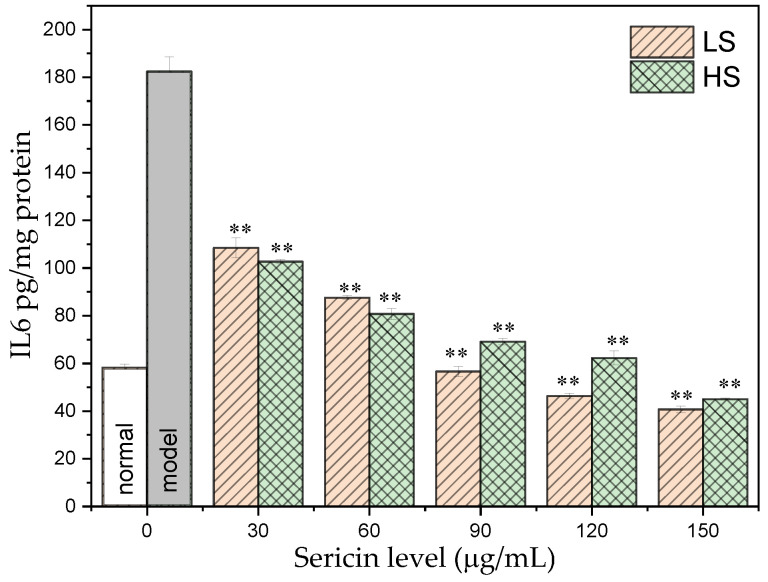
Effect of sericin peptides on IL-6 level in AML12 cultured with high glucose. ** *p* < 0.01, sample group vs. model group.

**Figure 5 antioxidants-11-00712-f005:**
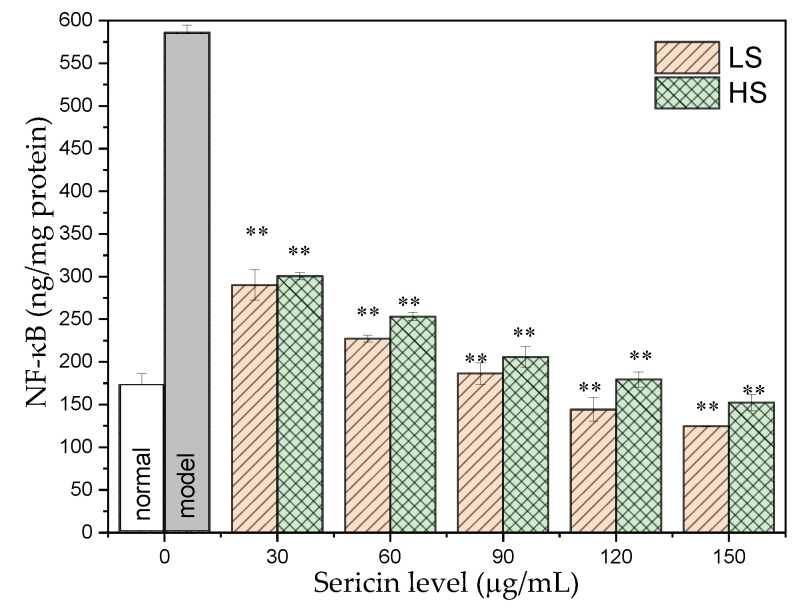
Effect of sericin peptides on NF-κB content in AML12 cultured with high glucose. ** *p* < 0.01, sample group vs. model group.

## Data Availability

Code and material; The datasets used and/or analyzed during the current study as well as analysis scripts are available from the corresponding author on reasonable request.

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
