# Peer review of "The Protective Effect of Sericin on AML12 Cells Exposed to Oxidative Stress Damage in a High-Glucose Environment"

_antioxidants, 2022, doi:10.3390/antiox11040712_

Round 1

Reviewer 1 Report

The article has been improved after the first revision. I can accept it in this form.

Author Response

English language and style  have been corrected and revised.

Reviewer 2 Report

Jing et al. studied the antioxidant and anti-inflammatory effects of silk sericin peptides on murine hepatic cell line AML12, cultured in high-glucose medium.  Both low (LS) and high (HS) molecular weight peptide preparations were shown to exert dose-dependent effects, although LS peptides were more effective. These findings are potentially very interesting as they suggest a green utilization of silkworm cocoon shells waste products obtained during silk processing. Unfortunately, this paper, even though revised, still does not meet the satisfactory standards. The main issue is that the authors addressed the biological problems, such as inflammation without a sufficient understanding of the underlying mechanisms. Thus, the conclusions formulated by the authors are based on insufficient evidence. In addition, some experiments are inadequately described.

Major points:

  1. The anti-inflammatory effect of sericin peptides were assessed by a number of Elisa-based methods, but none of these assays is adequately described i.e. the catalog numbers of specific kits are still missing and the indicated manufacturer company does not provide any info on Elisa kits.
  2. The main problem is that the authors claim that they used these ELISA kits to accurately quantify the concentration of TNF-a, IL-6 and NFkB in cell extracts obtained by freeze and thawing of cells. The ELISA assays have been widely validated by the scientific community for the detection of secreted cytokines in conditioned media and there are not enough data on reproducibility and reliability of ELISA kits applied on cell lysates. In addition, the authors performed the assays on unstimulated hepatic AML cells, which are known to produce very low levels of cytokines. The low levels of intracellular cytokines is a major methodological challenge, especially if unstimulated hepatocyte-derived cells are assessed. Therefore, the authors should provide more detailed description of the assays used. They should also indicate other authors who successfully applied this method to measure the inflammatory status of a cell culture like AML by providing specific literature references.
  3. Finally, the authors do not seem to understand the role and mechanisms of action of NFkappaB in inflammation. They looked at the downregulation of “nfkB content”, but t is not clear which element of NFkB pathway was analyzed. I looked for hours for other examples of the use of Elisa kits for the detection of “NFkb expression levels” and I did not find any papers.

The authors should demonstrate NFkB activity downregulation by sericin peptides by additional experiments, such as: 1) Western blotting of IkappaB degradation kinetics or 2) western blotting of nuclear and cytoplasmic cell lysate extracts to prove the translocation of one subunit of NFkb complex to nucleus upon sericin peptide treatment, or 3) any functional NFkB reporter assay measuring NFkB activity.

  1. The production of LS and HS peptides has been described in a recent paper in Biomolecules by the authors, therefore the method description should be adequately shortened and referred to in Materials and methods and not in the discussion. The authors should show the data or at least describe how the quality of sericin peptide preparations was assessed.

Minor problems:

  1. The introduction and discussion sections should be improved as they are quite chaotic. 
  2. It is not clear what post-test method was applied for ANOVA analysis.
  3. The manuscript contains also many other minor problems.

Author Response

Dear Dr. Editor,

I submit my revision manuscript (1639914) entitled “The protective effect of sericin on AML12 cells exposed to oxidative stress damage in a high-glucose environment” to Journal antioxidants. According your suggestion and Reviewer-2’s comments, we have carefully corrected and modified the related contents marked with blue word in revised manuscript. The detailed response list and revised manuscript was attached below. Thank you very much!

With best regards!

Zhang, Yu-Qing PhD/Prof.

Soochow University

March 22, 2022

Reviewer 3 Report

Authors significantly improve the manuscript quality.

Author Response

English language and style have been corrected and modified.

This manuscript is a resubmission of an earlier submission. The following is a list of the peer review reports and author responses from that submission.

Round 1

Reviewer 1 Report

The article is interesting. These studies follow the trend of using by-products and waste products for other purposes, which significantly reduces the burden on the environment and may have benefits, including health benefits.

The article requires a few corrections.

1.) Lines 40-43: Some text is written in superscript. Please amend it to plain text.

2.) The reference numbering in the text and in the list of references is written in a different style. Please correct the numbering in the text to the correct one (1, 2, 3 ...).

3.) Materials and Methods: Apart from the type of research, measurement and control equipment, the unit models and all the data of their manufacturers should be provided, i.e. name, city, state (if necessary, e.g. in the case of the USA), country. The authors did not mention the cities of the apparatus manufacturers. Please complete these details.

4.) Lines 98-99, 108, 139: Please recalculate the values of centrifugation speed and enter them instead of "rpm" in "g" unit.

5.) Chapters 2.4 and 2.6: No references to determinations. Have the determinations been made according to own methodology, methodology of other researchers or modified methodology of other researchers?

6.) Chapter 2.5: No references to determinations. Was the determination performed only on the basis of the instructions of the ROS determination kit? Have the determinations been made according to own methodology, methodology of other researchers or modified methodology of other researchers?

Reviewer 2 Report

Jing et al. studied the antioxidant and anti-inflammatory effects of silk siericin peptides obtained by two alternative ultrasonic degradation methods, on murine hepatocyte-derived cell line AML12, cultured in high-glucose medium. Both low (LS) and high (HS) molecular weight peptide preparations were shown to exert dose-dependent effects, but LS peptides were more effective. These findings are potentially very interesting and may suggest a green utilization of waste produced during silk processing for production of silk based functional foods. Nevertheless, it is not clear how some experiments presented in this paper were conducted, as experimental details are totally missing, while other experiments are inadequately described. Thus, the conclusions formulated by the authors are based on insufficient evidence.

Major points:

  1. No chemical assessment method is provided to characterize the size of peptides in LS and HS preparations. The authors described the preparation of peptides with sufficient detail, but the analysis of peptides present in these two different siericin formulations is missing.
  2. The antioxidant effects of LS and HS are assayed by one inadequately described method. It is not clear what kind of ROS are measured and how they are measured. The authors report only the name of the manufacturer of the kit used to perform the assay, but the chemical principle is not clear. Beside that, the antioxidant effect of silk-derived peptides should be verified by at least two alternative assays to provide fully convincing data.
  3. The anti-inflammatory effect of siericin peptides was assessed by a number of Elisa-based methods, but none of these assays is adequately describede. the catalog numbers of specific kits are missing and the indicated manufacturer company does not provide any info on Elisa kits. Therefore, there is not enough information to assess the reliability of these tools.
  4. The main problem is that the authors claim that they used these ELISA kits to accurately quantify the concentration of TNF-a, IL-6 and NfkB in cell extracts obtained by freeze and thawing of cells. The ELISA assays have been widely validated by the scientific community for the detection of secreted cytokines in conditioned media and there are not enough data on reproducibility and reliability of ELISA kits applied on cell lysates. In addition, the authors performed the assays on unstimulated hepatic AML cells, which are known to produce very low levels of cytokines. The low levels of intracellular cytokines is a major methodological challenge, especially if unstimulated hepatocyte-derived cells are assessed. Therefore, the authors should provide more detailed description of the assays used. They should also indicate other authors who successfully applied this method to measure the inflammatory status of a cell culture like AML by providing specific literature references.
  5. Finally, the authors should confirm at least the downregulation of “nfkB content” in HS- and LS-treated AML cells by Western blotting of nuclear AML cell lysates for a specific subunit of NFkb complex or any functional NfkB assay measuring NfkB activity.
  6. All figures in the manuscript lack fully explanatory legends and statistical data.
  7. The Ms contains also many minor problems. For example, it is not sufficiently referenced; the citations are indicated by Roman numbers; the introduction and discussion sections should be improved as they are quite chaotic. 

Reviewer 3 Report

In this article, the authors evaluated the protective effect of two different silk sericins againt the oxidative stress damages induced on murine hepatocytes.

I suggest some modifications and integrations.

Comments:

  • Line 38-43: There are formatting problems. Please modified.
  • The number used for literature citation in the main text and in the References section are not the same (Roman vs Arabic numbers): please use the same formatting.
  • Do you evaluated the Ca(OH)2 residues contained in the LS product? Do you evaluate the possibility to purify the solution using a ultrafiltration or a dialysis method? When you add the sulfuric acid which is the final obtained pH?
  • Page 3, line 122 and page 3, line 128: “appropriate amount of samples”: please specify the added volume and concentration.
  • Regarding the cell viability, do you consider the same high-glucose culture medium? Please specify all culture conditions.
  • Figure 1 and figure 2: please specify “sericin concentration” in x-axis.
  • Please indicates the statistical differences between the groups in all figures.
  • Could be interesting to discuss the higher protective effect of LS with respect to HS. Do you hypothesize why LS is more effective? Please discuss this aspect that could be more interesting for readers.
  • Conclusion section repeats the results yet reported in the paper. I suggest to add the future perspective, the limits and the advantages of the use of sericin. Do you evaluate the feasibility of a large-scale production of the proposed product? Do you evaluate to use not only a silk sericin derived from cocoons but also from wastewater obtained in textile industry? Recently some authors (Wu et al., Food Chem 2007, DOI: 10.1016/j.foodchem.2006.10.042; Orlandi et al., J Chem Technol Biotechnol 2020, DOI:10.1002/jctb.6441) proposed this approach and I think that could be very interesting for recycling purposes.